# Study of the Microstructure of Amorphous Silica Nanostructures Using High-Resolution Electron Microscopy, Electron Energy Loss Spectroscopy, X-ray Powder Diffraction, and Electron Pair Distribution Function

**DOI:** 10.3390/ma13194393

**Published:** 2020-10-01

**Authors:** Lahcen Khouchaf, Khalid Boulahya, Partha Pratim Das, Stavros Nicolopoulos, Viktória Kovács Kis, János L. Lábár

**Affiliations:** 1École Nationale Supérieure des Mines-Télécom de Lille-Douai Lille Douai, Lille Université, CEDEX, 59653 Villeneuve D’Ascq, France; lahcen.khouchaf@imt-lille-douai.fr; 2Departamento de Química Inorgánica, Facultad de Qúimicas, Universidad Complutense, 28040 Madrid, Spain; khalid@quim.ucm.es; 3Electron Crystallography Solutions SL, Calle Orense 8, 28020 Madrid, Spain; 4NanoMEGAS SPRL, Blvd Edmond Machtens 79, B-1080 Brussels, Belgium; 5Institute of Technical Physics and Materials Science, Centre for Energy Research, 1121 Budapest, Hungary; kis.viktoria@energia.mta.hu (V.K.K.); labar.janos@energia.mta.hu (J.L.L.)

**Keywords:** amorphous silica, powder diffraction, transmission electron microscopy, high-resolution, spectroscopy, electron diffraction, electron pair distribution function

## Abstract

Silica has many industrial (i.e., glass formers) and scientific applications. The understanding and prediction of the interesting properties of such materials are dependent on the knowledge of detailed atomic structures. In this work, amorphous silica subjected to an accelerated alkali silica reaction (ASR) was recorded at different time intervals so as to follow the evolution of the structure by means of high-resolution transmission electron microscopy (HRTEM), electron energy loss spectroscopy (EELS), and electron pair distribution function (e-PDF), combined with X-ray powder diffraction (XRPD). An increase in the size of the amorphous silica nanostructures and nanopores was observed by HRTEM, which was accompanied by the possible formation of Si–OH surface species. All of the studied samples were found to be amorphous, as observed by HRTEM, a fact that was also confirmed by XRPD and e-PDF analysis. A broad diffuse peak observed in the XRPD pattern showed a shift toward higher angles following the higher reaction times of the ASR-treated material. A comparison of the EELS spectra revealed varying spectral features in the peak edges with different reaction times due to the interaction evolution between oxygen and the silicon and OH ions. Solid-state nuclear magnetic resonance (NMR) was also used to elucidate the silica nanostructures.

## 1. Introduction 

SiO_2_ (silica) is a three-dimensional siloxane bridged bond structure that is used as an aggregate or in nanocrystalline form, which, in recent decades, has been widely employed to create high-performance or highly functional materials [1]. The use of silica as an aggregate in silica glass has been intensively investigated for its specific heat insulation, good optical transmission, and high chemical resistance properties. Amorphous silica (a-silica) has various industrial (i.e., glass formers) and scientific applications, such as in photovoltaic cells and in electronic devices with optical properties [2,3,4]. Determining the response of porous silica to densification is challenging, as some amorphous materials are known to display anomalous behavior under high pressure [5]. Natural silica is used as an aggregate in composite materials such as concrete [6,7,8]. In fact, the degradation of concrete depends on the crystalline quality of the aggregate, where the reactivity of silica is dependent upon the chemical process that occurs between the amorphous or poorly crystallized silica present in mineral aggregates, referred to as an alkali silica reaction (ASR) [9,10,11]. 

Likewise, nanosilica is used in pottery clay materials as a strengthening additive, in electronic compounds as an insulator, and in the glass industry. It is also used to improve the creep resistance of thermoplastic polymers [12,13,14,15]. In contrast, the use of nanopowders presents a major health hazard, either during the manufacturing process or following the wear of a material, which may cause the release of nanoparticles into the environment. To solve this problem, it is of interest to manufacture nanostructured silica (<100 nm) in the form of arranged clusters. The advantage of this process is the low dimensionality properties of silica, while avoiding health risks during the manufacturing process following the wear of the material and its subsequent release into the environment. In addition to its technological relevance, the availability of amorphous nanostructured silica materials will enable the study of a vast range of interfacial phenomena and confined species.

Many previous studies have shown that the reactivity of silica compounds is due to the amorphous and strongly disordered part of silica. Different works have been carried out in order to characterize the nano- and micro-structures of concrete so as to improve its durability [16,17,18]. Micro-X-ray absorption near-edge structure (XANES) and micro-fluorescence experiments have been carried out to investigate the local structural evolutions of a heterogeneous and natural silica submitted to the ASR process [8]. Using micro-beam sources, micro-zones with different properties have been studied. Elemental maps obtained by environmental scanning transmission electron microscopy (ESEM) and micro-X-ray fluorescence (micro-XRF) demonstrate the accurate diffusion of potassium inside grains. Using Si K-edge XANES spectra has enabled the elucidation of the structural evolution induced by the alkali–silica reaction in silica from the outside to the inside of particles, showing no significant changes in the K cations [8,18,19,20]. However, some questions remain unanswered, for instance, regarding the contribution of different structural forms (i.e., amorphous and disordered) of silica in the ASR process.

Previous ASR studies on natural flint [8,18,19,20,21] have shown that the reaction begins with breaking the Si–O–Si bonds of the siloxane bridge and the formation of amorphous and nanocrystalline phases. However, the structural heterogeneity of flint, i.e., the presence of microcrystalline, nanocrystalline, and amorphous domains, complicates the study of degradation mechanisms and the reaction kinetics. All previously published studies have proposed incomplete structural models [16,17,18,19,20,21]. 

To explore and predict the properties and interfacial behavior of silicon, silica, and its hydrolysis, Van Din et al. [22,23] developed the “ReaxFF” reactive force field computational tool. In general, ReaxFF describes the breakdown and formation of bonds due to calculations of bonding states using interatomic distances [24]. The developed force field is empirical and bond order-dependent and requires fewer computer resources in comparison to methods based on quantum mechanics. The parameters of this force field were recently further developed to describe correctly the O migration mechanism in an Si network [25] at different temperatures (i.e., 880–2400 K).

The structural study of nanocrystalline amorphous materials is typically performed by neutron or X-ray diffraction [26]. However, the scattering cross-section of electrons is relatively large compared to neutrons or X-rays, making it easier to study nanovolumes using transmission electron microscope (TEM). The use of the electron pair distribution function (e-PDF) in TEM is ideal for studying amorphous materials where the acquisition time of e-PDF spectra is much shorter (milliseconds instead of hours) compared to neutron or X-ray diffraction. In contrast, the evolution of structures using a small quantity of a sample is more easily studied using electron diffraction and e-PDF analysis [27,28]. Currently, e-PDF is used only to extract short–long-range order information from nanoparticles, amorphous thin films, and amorphous organic materials. At present, there is not any work in the scientific bibliography related to extracting structural information on studying chemical reactions using e-PDF, which is an ideal method for studying structural changes in nanovolumes.

In this work, for the first time, we used a combination of high-resolution transmission electron microscopy (HRTEM), X-ray powder diffraction (XRPD), electron energy loss spectroscopy (EELS), electron pair distribution function (e-PDF), and solid-state nuclear magnetic resonance (NMR) to study the structural evolution of silica nanostructures with different reaction times during the ASR process. By combining different techniques, we were able to consistently correlate the morphology changes of amorphous silica nanostructures (information from HRTEM) with relation to Si-O environmental changes (information from EELS and e-PDF); in addition, we also confirmed those structural changes in bulk silica using XRPD. Short-range ordering (SRO) using e-PDF was also observed in the material even after several hours of hydrothermal reaction. In parallel, formation of Si-OH on the surface was also confirmed by NMR. In the following sections, we will present and discuss in detail our results related to the combined use of previously mentioned techniques.

## 2. Materials and Methods

The starting material used in this study was a-silica from Alfa Aesar (www.Alfa.com) (Ward Hill, MA, USA). The purity of silica was confirmed to be 99.9% by X-ray fluorescence analysis. The a-silica was submitted to the ASR process as previously described [20,21]. Briefly, 1 g of a-silica (S1) was submitted to an accelerated ASR process at 80 °C for 6 hours (S2), 168 hours (S3), and 312 hours (S4), with a mixture of 0.5 g of portlandite Ca(OH)_2_ and 10 mL of potash solution KOH at 0.79 mol/L. The sample was retained under the ASR process for different periods in order to track the evolution of the resulting structure. Calcium and potassium were then removed by a selective acid treatment [20,21].

The XRPD spectra were recorded for 2θ values between 5° and 60° with steps of 0.007° and a counting time of 10 s per step using a Bruker D8 ADVANCE (Bruker AXS, Karlsruhe, Germany) diffractometer operating at 40 kV and 40 mA with Cu radiation (λCu = 0.15418 nm). 

HRTEM and EELS measurements were carried out on a JEOL 3000FEG transmission electron microscope operating at 300 kV, equipped with a Gatan Enfina EELS spectrometer at the Electron Microscopy Centre, Madrid, Spain. The EELS energy resolution was approximately 1.2 eV for all spectra, as measured by the full-width at half-maximum (FWHM) of the corresponding zero-loss peak. Both the background and the plural scattering had to be subtracted from the experimental spectra to isolate the white line intensities. All analyzed crystals were very thin nanoparticles (mean free path λ ≤ 1), where the EELS spectra were based on the fine edge of single nanoparticles to reduce the influence of multiple scattering effects.

For the electron diffraction (ED) measurements, the silica particles were dispersed onto the surface of carbon foil (20–30 nm thickness) that covered the Cu TEM grid (Tedpella, Redding, CA, USA). The ED measurements were performed using the nanoprobe mode of a Philips CM20 TEM (Philips Electron Optics division, Amsterdam, Netherlands) with an LaB6 cathode operating at 200 keV. The ED patterns were recorded on imaging plates (Ditabis), which provide a linear response to an electron dose over 6 orders of magnitude. We used the TEM nanoprobe mode (where the diameters of the studied and illuminated areas coincide) so as to avoid stray radiation from areas outside of the selected area (SA). This is essential in the e-PDF analysis of amorphous materials, as the signal from the amorphous structure is very weak—practically of the same order of magnitude as background—and therefore, even a small variation in intensity could greatly affect the result of the measurements. It is well known that nanocrystalline silica and hydrated silicates are highly sensitive to radiation, which undergo amorphization under an electron beam in a matter of seconds. To minimize beam damage [29], the ED patterns were obtained at a liquid nitrogen temperature and the incident electron beam intensity was kept as low as possible. The electron dose rate during the measurement was kept at the level of 10 e-/Å^2^ s, which is far below the critical dose for this material. 

The standard procedure we used to obtain reproducible ED patterns was as follows: (a) Start with the de-magnetization of all lenses, (b) set a fixed current for the objective lens, (c) position the studied area to the focal plane of the objective lens (using Z-control), and (d) set fixed values for the condenser lenses that focus the diffraction pattern [30,31,32]. For camera length calibration, self-supporting nanocrystalline Ni foil was used. Using this standard procedure, deviations of the camera length could be kept below 0.5%, regardless of the different samples examined. During the ED experiments, the beam diameter was set to ca. 1.5 µm, and for optimal background subtraction after recording the ED patterns from the silica particles, an additional ED pattern was recorded from the empty carbon foil under identical beam conditions. Radial averaging of the ED patterns was performed using process diffraction and the e-PDFSuite software [33,34,35,36]. The ED intensities were integrated radially into one-dimensional (1D) profiles and the background intensity from the carbon foil was subtracted from that of the silica particles. The resulting 1D ED profile, which contained intensity scattered exclusively by the silica particles, yielded input data for the e-PDF analysis. 

Solid-state ^29^Si magic angle spinning MAS NMR experiments were carried out on the Bruker Advance 100 spectrometer (Bruker BioSpin, Billerica, MA, USA) operating at magnetic fields of 2.34 T. The ^29^Si MAS experiments, operated at 19.89 MHz, and the samples were spun in a cylindrical 7 mm ZrO_2_ rotor at a spinning frequency of 4 kHz. ^29^Si chemical shifts were determined relative to the tetramethylsilane as an external reference. The spectra were recorded with a pulse angle of π/5 and a recycle delay of 80 s, which was verified to enable relaxation. For each sample, a total of 256 scans were carried out.

## 3. Results and Discussion

Structural silica has SiO_4_ tetrahedral units connected at the corners by bridging O. The ideal silica two-dimensional (2D) structures have linear Si–O–Si bonds, as illustrated in Figure 1, where the torsional energy required for one bond rotation depends on the bond angles that vary between 145° and 150° [37,38]. In the literature, the nature of Si–O–Si angles is extensively debated within the glass community, and an overview of the literature regarding measured and simulated Si–O–Si angles can be found in Reference [39]. Due to the large variety of Si–O–Si angles that join two neighboring building units, a-silica structures lack long-range order. 

Thereby, the local disorder and the orientation of SiO_4_ tetrahedra could allow Si–O–Si angles to adjust to values that are more chemically stable. On a short-length scale, different phases may experience similar dynamic disorder regarding atom positions. In a recent study, we confirmed the hypothesis of the formation of silica clusters of Si–O with different structural states following the ASR process [40]. Depolymerization of the silica network creates Q4 (Si(OSi)_4_) species, Q3 (Si(OSi)_3_)(OH)) species that consist of a tetrahedral silicate sheet structure, Q2 species (SiO_4_ tetrahedra in the middle of silicate chains), and Q1 species (tetrahedra at the end of silicates chain), as confirmed in a previous study [41]. Therefore, the degradation results of our material are not based on a simple formation of amorphous phases, but rather on a formation of nanodomains of heterogeneous sizes with different structures. In contrast, no high spatial resolution information exists about the electronic and chemical environment around silicon and oxygen within these structures.

The high-resolution electron microscopy (HREM) analysis of the samples was performed under the same magnification and the same defocus to better understand the microstructures of the amorphous state. Figure 2 shows the obtained micrographs studied for all compounds (i.e., S1 (a), S2 (b), S3 (c), and S4 (d)). We observed that two compounds (i.e., S1 and S2) were very stable under the beam, while the other two (i.e., S3 and S4) were slightly unstable after ~1 min of TEM. To avoid any artefact in the HRTEM images due to beam damage, data from each sample were collected under the same electron dose condition, far below the critical dose of the material. The images corresponding to samples S1 and S2 showed an amorphous microstructure, while those of S3 and S4 showed small changes compared to S1 and S2, since the presence of Si–Si ordering in a SRO with different arrangements of the Si–Si domains was observed (see enlarged images inserted in Figure 2c,d).

All of the studied materials were found to be amorphous (Figure 2), which was confirmed also using XRPD (Figure 3) and subsequent e-PDF analysis (Figure 8). The HRTEM micrograph in Figure 2a shows the morphology of the starting amorphous silica. The evolution of the silica ring formation network can be observed in S3 and S4, and in Figure 2b–d, the morphology of the amorphous silica can be observed after the reaction. An increase in the size of the nanopore ring formation marked with white dots in the inserts of Figure 2c, d in the silica network can be more clearly observed for S4 than for S3. In addition, the nanopore distribution is highly heterogeneous in S3 and S4, leading to the formation of short-range ordering only. As an important remark, it is interesting to note that the number of Si atoms that form part of the Si tetrahedral ring is lower in S3 than in S4, as indicated by the white dots in Figure 2c,d (magnified insert).

In Figure 3, the XRPD pattern of amorphous silica and subsequent reaction materials shows typical diffuse peaks, confirming the presence of an amorphous structure and/or very small crystal sizes [42]. For the starting material (S1), a broad diffuse peak maxima is located at 2θ = 21°, which is a well-known feature for amorphous silica material. A subsequent shift of the maximum is observed toward the higher angles with increasing ASR reaction times.

As observed during the HRTEM image acquisition, the particles are of nanometer scale, producing broad Bragg peaks in the XRPD data for the studied materials. With the ASR process, it is possible that the Q4 species convert to Q3, Q2, and Q1 species. As the reaction time increases, the Q1 and Q2 species may form, creating increased diversity in the silica structure bond distance distribution, with broader diffuse peaks in the XRPD data. 

With higher reaction times, the average bond distance in the structure becomes shorter, possibly due to more O–H species connected to Si, which translates into a shift of the X-ray diffraction peaks toward higher angles, as observed in the XRPD data for S4. The change in structure, as observed in the XRPD patterns, is consistent with the HRTEM images.

To shed more light on the structural changes, EELS analysis was performed to observe variations in the detailed features around the Si–L and O–K edges. The obtained EELS spectra were compared with EELS reference bibliography data [43] to check for possible energy shifts or any variation in the shape of the edges.

It is believed that the incorporation of OH^-^ groups into the amorphous silica system generates a mixed Si electronic configuration. Therefore, for atoms such as Si, the L_2,3_ ratio might be expected to vary following electronic configuration changes. To establish the formal electronic configuration and/or oxidation state of Si in each composition/reaction time, EELS analysis was performed by comparing the evolution of the Si L_2,3_ edges for the whole studied range/reaction times (S1–S4), as shown in Figure 4. 

The EELS spectra corresponding to S1–S4 are shown in Figure 4a. The Si *L*_2,3_ edges for the studied compounds were normalized to the L3 maximum intensity. The energy loss near-edge spectra (ELNES) show a first defined peak around 107 eV (marked a), followed by a broad peak at 114 eV (marked b), and then a very broad peak with a maximum at 130 eV (marked c). The results show that although Si–L_2,3_ core loss edges have an appropriate energy loss range (approximately 108 eV) [43], and the peaks are located at similar positions in the four spectra, the relative peak intensities change dramatically from S1 to S4. Although Si peaks can be observed for all samples, they change from broad (S1) to sharp (S4) peak positions as the reaction time increases. Such a difference in the Si–L_2,3_ peaks is probably due to Si tetrahedral distortion, since in the SiO_4_ tetrahedra, the four distances between the silicon and the oxygens are slightly different. The higher reaction times indicate that the OH^−^ ions were incorporated into the silica amorphous structure, which may be associated with the increase in the white line Si–L_2,3_ intensity, and generally, the spectra become better defined with sharper EELS peaks. The appearance of a sharper EELS peak as the reaction time increases is probably due to the incorporation of OH^−^ around Si atoms, leading to more regular (less distorted) Si tetrahedra [43]. A small shoulder observed in S4 indicates that some of the Si tetrahedra remain distorted in the measured sample, since the perfect maxima are only observed if all of the Si tetrahedra are undistorted [44]. The EELS spectra show strong modifications of the silica tetrahedral network at different reaction times. The ratio between peaks a to b deduced from the Si–L_2,3_ EELS are shown in Figure 4b, highlighting that instead of a linear increase, hydration happens very fast with a huge distortion in the Si tetrahedra in the first instance, and then a tendency towards plateaus.

We also studied the evolution of the O–K edges of the EELS spectra with different reaction sample species (S1–S4) to provide information on the coordination structure of local oxygen atoms, such as the configuration and the type of neighboring species. The O–K edges in the spectrum were caused by the transition from the O 1s state to the O 2p final state in the conduction band, hybridized with the valence orbitals of neighboring atoms. The features of the O–K edges of all compounds were found to be quite similar (Figure 5). 

Generally, the sharpness of peak “a” of the O–K edges arose from O–O scattering and increased in intensity (Figure 5a) as the number of O second-nearest neighbors around the excited O atom increased [39]. The results show that although the O–K edges have an appropriate energy loss range (approximately 532 eV) [45,46] with peaks located at similar positions in the four spectra, the relative intensity of the peaks changes from S1 to S4 with different reaction times, where the observed increase in the O–K peak intensity is related to the presence of additional oxygen atoms around the excited oxygen atoms. The EELS spectra therefore show strong modifications regarding the oxygen environment at different reaction scales from samples S1 to S4. The ratio between peaks a to b deduced from the O–K EELS spectra (Figure 5b) shows a tendency towards a plateau instead of a linear increase with reaction time, which can also be observed in the EELS spectra of Si (Figure 4b).

Such an increase could be correlated with an accelerated hydrolysis reaction of the OH^−^ groups with silica from S1 to S4, in agreement with the infrared (IR) spectroscopy results published by Hamoudi et al. [47]. Figure 6 provides a schematic of the amorphous silica–OH^−^ interaction, where the reaction mechanism starts at the surface and, via a stepwise mechanism, hydrates the silica, in a similar way as proposed previously by Dove et al. [48] in the case of quartz–water interactions. 

To complete the results obtained by HRTEM, EELS, and XRPD, a ^29^Si MAS-NMR experiment was performed in order to obtain information about short-range order changes and about the surface of silica nanostructures. NMR experiments were performed only for two samples with very different behaviors in order to check the creation of Q3 species. Figure 7 shows the ^29^Si NMR-MAS corresponding to the starting form of silica (S1) and silica after a reaction time of 312 hours (S4). In the starting form of silica (S1), the major species are Q4, corresponding to amorphous silica with SiO_4_ tetrahedral units, each one connected to four tetrahedra via oxygen. The spectrum for S1 presents a line, centered around –110 ppm, that is attributed to Q_4_ species, which has also been confirmed by previous studies [49,50,51]. 

For the final reaction material, S4, in addition to Q4 species, a second line centered around –101 ppm can be observed, which can be attributed to the Q3 species that correspond to silanol groups Si–OH. The formation and changes of the Si–OH species and Si–O–Si bonds were also confirmed previously in a similar material using Fourier transform infrared spectroscopy [47]

From the collected ED data (samples S1–S4), the electron pair distribution function (G(*r*)), which provides a measure of the probability of finding two atoms separated by distance *r* (Figure 8), was calculated using the e-PDFSuite and the Process Diffraction software [34,35,36,52] developed to analyze the ED patterns of amorphous and nanocrystalline materials. During the calculation of the 1D distribution from the 2D electron diffraction patterns of silica materials, the contribution of amorphous carbon support was subtracted where the distortions in the 2D diffraction patterns were corrected. The beam stopper area and the dead pixels in the detector were masked and eliminated from the ED pattern during e-PDF calculations. The detailed procedure of the G(*r*) calculation from the 1D distribution data using the e-PDFSuite and Process Diffraction software program has been described elsewhere [34,35,36]. 

Figure 8a shows that, following the e-PDF analysis, no peak (corresponding to interatomic distances) can be found beyond 8 Å in any of the hydrothermal products (S1–S4). This fact confirms that only short-range ordering is present in the material, even after several hours of hydrothermal reaction. It is interesting to note that the minor ripples observed in the e-PDF results beyond 8 Å are due to the limited Q range resolution and the attenuation of the signal amplitude (obtained Q = 14 Å^−1^, where Q = 2*Π/*d), and they do not contain any structural information. In the 1.3–3.5 Å region for S1–S4, small changes can be observed in the e-PDF peak positions (corresponding to the interatomic Si–Si, Si–O, and O–O distances). The small peak at 2.0 Å is probably an artifact, due to limited Q resolution or due to additional Si–O connectivity in the amorphous state. All such peaks/interatomic distances in the 1.3–3.5 Å region match well with the distances that exist within the silica crystalline structure [53], as shown in Figure 8b. As a consequence of the precision of the peak localization in the e-PDF analysis decreasing with increasing atomic distances due to the enhanced peak broadening, the relatively sharp first, second, and third e-PDF peaks corresponding to the nearest interatomic distances can be reliably considered for our comparison (Figure 8b). According to the results, small changes (i.e., shortening of bond distances) can be observed for species with higher reaction times (more obvious for S4 in comparison to S1), which confirm our previous observation from the XRPD patterns and EELS spectra of the Si and O edges. 

The interatomic distances extracted from the e-PDF calculation are accurate, as the ED pattern calibration and the determination of the center of the ED pattern may cause an error smaller than ± 0.005 Å within the range of 1–5 Å [54,55,56]. To establish high accuracy of the observed peak positions, during our study, a proper calibration of the TEM camera length was carried out, and all the experiments were performed under similar microscope conditions. Electron diffraction is subject to multiple scattering, and this multiple scattering is prominent for nanomaterials with higher particle sizes. However, it was demonstrated by Anstis et al. [57] that multiple scattering does not affect peak positions in G(*r*) if t/α ≤ 5 (where t is the sample thickness and α is the elastic mean free path), although it may affect coordination number determination. On the other hand, as a-silica consists of only light elements, we did not consider correcting for multiple scattering during the G(*r*) calculation.

## 4. Conclusions 

The study of amorphous silica and its hydrothermal reactions at different time intervals by using various complementary techniques—namely, HRTEM, XRPD, NMR, and e-PDF—revealed the amorphous nature of the nanostructure of the resulting material. All of the studied samples were found to be amorphous, as observed by HRTEM, a fact that was also confirmed by XRPD and e-PDF analysis.

The combined results of these techniques shed light on the structural changes that occur at different reaction times. A shift of the diffuse peak maxima toward higher angles in the XRPD patterns indicates shortening of the average bond distance, a fact that was also confirmed by e-PDF analysis. The changes (i.e., shortening of bond distances) for species with higher reaction times observed via e-PDF analysis were also confirmed by the XRPD and HRTEM results.

The observed structural changes might be due to the Si–O tetrahedral contraction with increasing reaction times, which were also observed in the EELS spectra and could be explained by the depolymerization of the amorphous structure and the presence of some OH species, as also confirmed by NMR spectroscopy. The NMR results also revealed the presence of Si–OH species, indicating the existence of a three-dimensional network of silica. For the final reaction product (S4), the solid-state NMR analysis showed that, in addition to Q4 species, a second line centered around –101 ppm appeared, which can be attributed to the Q3 species that appeared in the silica nanostructures, thus confirming the results obtained using the EELS technique. 

Again, our current results are in agreement with a recent study concerning the densification effects on porous silica using molecular dynamics, where the rearrangement of Si–O–Si is part of an important structural change effect during densification. In addition, densification has been shown to produce more rings in vitreous silica, which could be attributed to the effect of repolymerization [5]. 

Finally, this study also showed the importance of using many combined state-of-the-art techniques to study complex structural problems related to the reactivity of amorphous silica. RMC (Reverse Monte Carlo) modeling of the obtained e-PDF data are in progress in order to provide additional insight about the amorphous silica structure.

## Figures and Tables

**Figure 1 materials-13-04393-f001:**
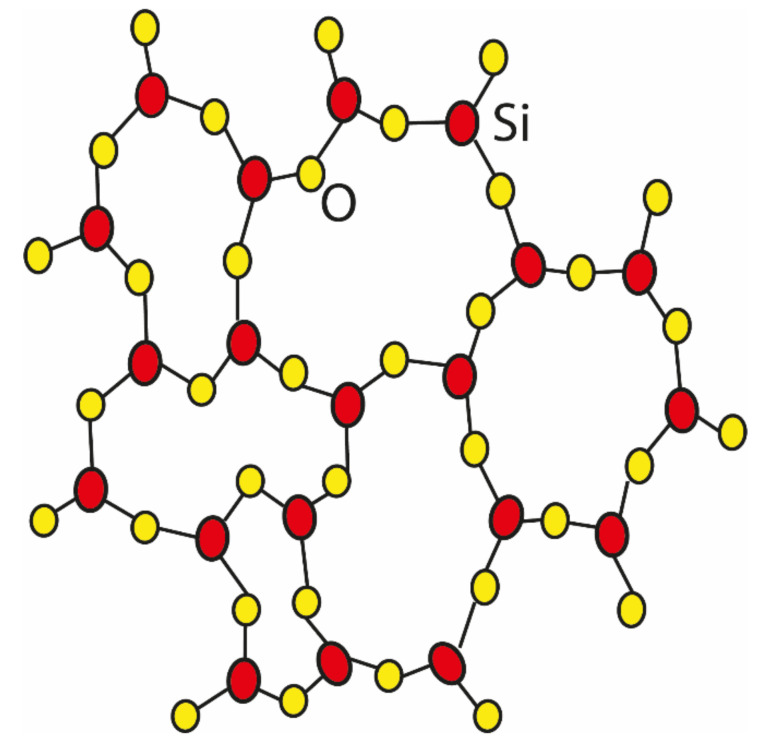
The ball-and-stick diagram of the structure of amorphous silica (adapted from Keen and Dove [33]).

**Figure 2 materials-13-04393-f002:**
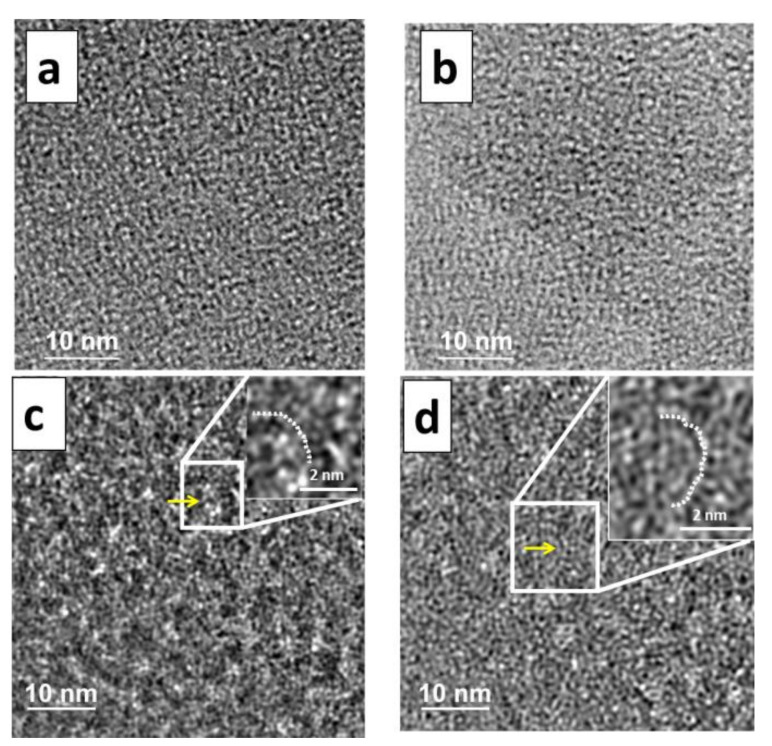
High-resolution electron microscopy images of the a-silica compounds S1 (**a**), S2 (**b**), S3 (**c**), and S4 (**d**). The area inside the white rectangle has been enlarged to show the ring structure. Enlarged images are inserted for S3 (**c**) and S4 (**d**), in which the nanopore ring formations are marked with white dots. Yellow arrows within the white square area indicate a nanopore ring formation.

**Figure 3 materials-13-04393-f003:**
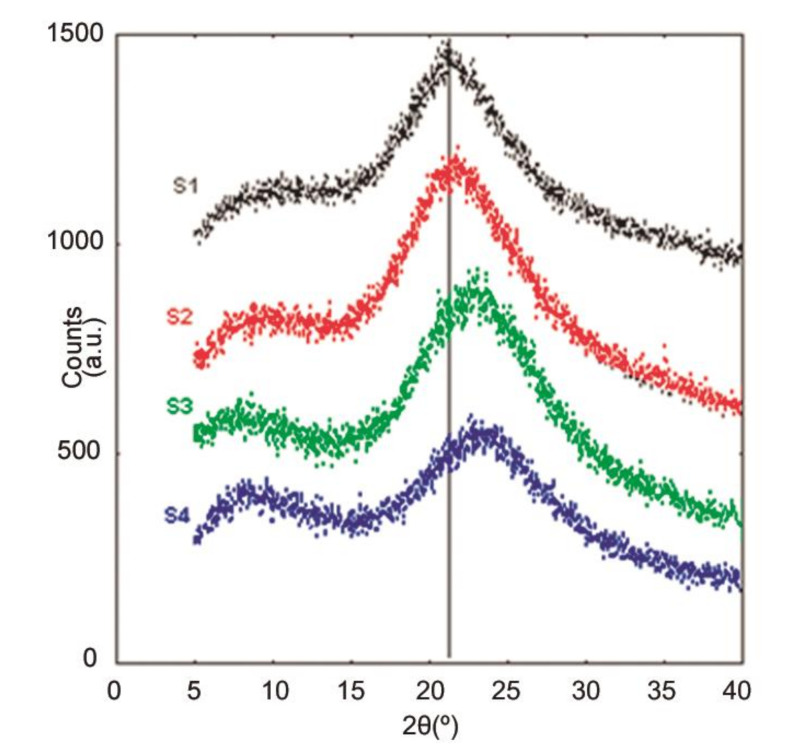
The X-ray powder diffraction (XRPD) spectra of a-silica (S1–S4) using various alkali silica reaction (ASR) times, where the x-axis is 2θ and the y-axis is the intensity in arbitrary units. The vertical line represents the peak maxima of the starting material.

**Figure 4 materials-13-04393-f004:**
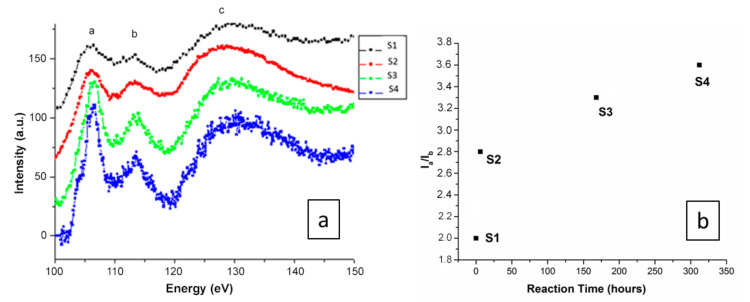
(**a**) Electron energy loss spectra of the Si L_2,3_ edges obtained for all compounds/reaction times. Data were normalized to the L_3_ maximum intensity. (**b**) Evolution of the intensity ratio of the peaks a to b deduced from the Si L_2,3_ electron energy loss spectroscopy (EELS) spectra.

**Figure 5 materials-13-04393-f005:**
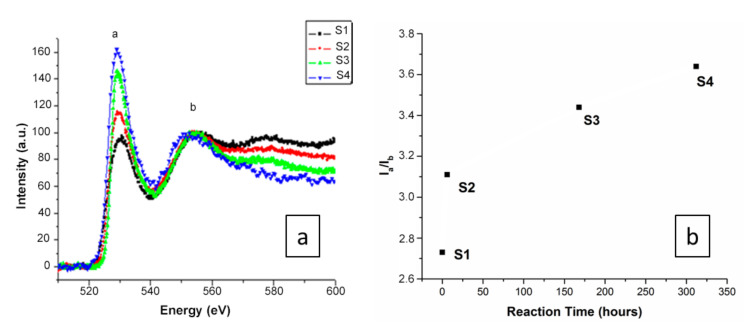
(**a**) Electron energy loss spectra taken of the O–K edges of all compounds. All of the spectra were calibrated at the O–K pre-edge peak position. (**b**) Evolution of the intensity ratio of peaks a to b deduced from the O–K electron energy loss spectroscopy (EELS) spectra.

**Figure 6 materials-13-04393-f006:**
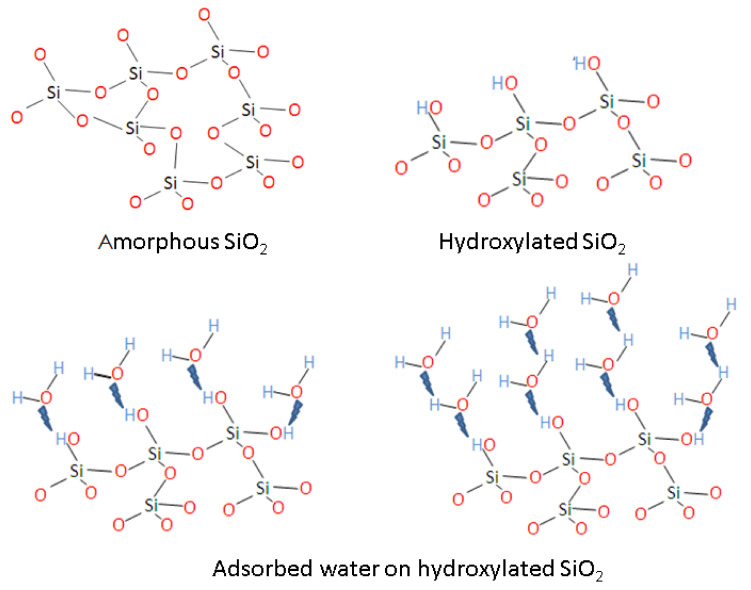
Schematic diagram illustrating stepwise OH^−^ interactions with a-silica. Hydrogen bonding between absorbed water molecules is not shown in the figure, as it is beyond the scope of our study.

**Figure 7 materials-13-04393-f007:**
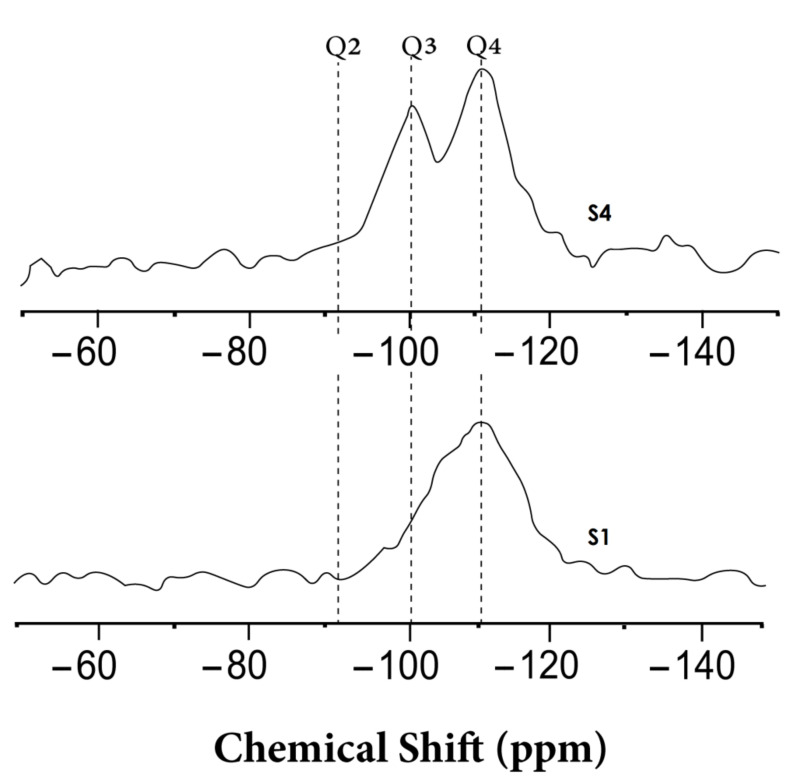
Local ordering of the starting form of a-silica S1 and after reaction time of 312 hours (S4): ^29^Si MAS NMR spectra, where the x-axis shows the chemical shift in parts per million (ppm) and the y-axis corresponds to the intensity scale of each spectral line in arbitrary units.

**Figure 8 materials-13-04393-f008:**
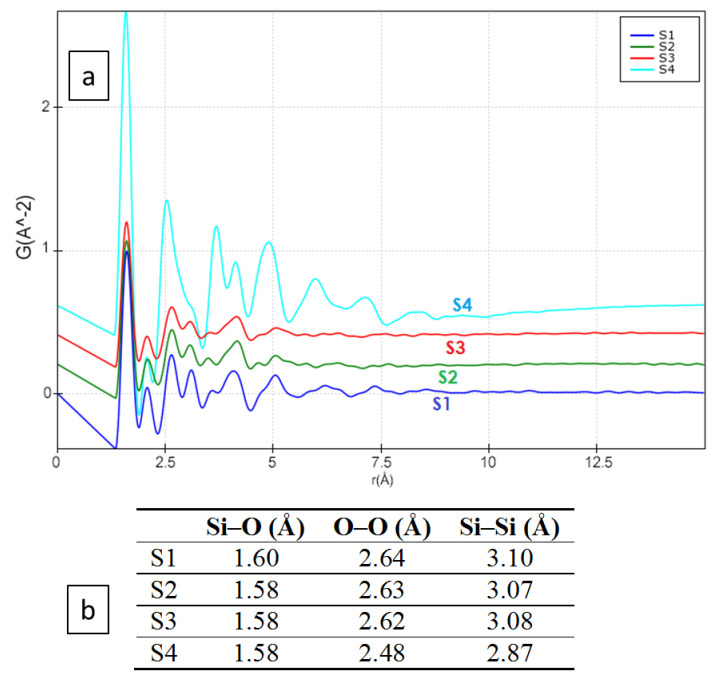
(**a**) The calculated electron pair distribution function (e-PDF) from S1–S4, where the x-axis represents the interatomic distances in Å and the y-axis represents the probability (in arbitrary units) of finding two atoms at distance *r*. (**b**) Calculated of the peak position from the e-PDF analysis.

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
