# Peer review of "Study of the Microstructure of Amorphous Silica Nanostructures Using High-Resolution Electron Microscopy, Electron Energy Loss Spectroscopy, X-ray Powder Diffraction, and Electron Pair Distribution Function"

_materials, 2020, doi:10.3390/ma13194393_

Round 1

Reviewer 1 Report

In this work, the evolution of amorphous Silica nanostructure was investigated using electron microcopy and electron pair distribution function. Overall, this work is well structured. I could recommend the publishment of this work after addressing the following major issues: 1. There are a lot of editing errors in this manuscript and the language is difficult to read. The language should be modified by a native speaker. 2. In the term of S4, a shoulder of peak a could be observed in Figure 4, which is obviously different from the other samples. Some explanations should be given.

Author Response

Dear Reviewer,

Thanks for your comments and suggestions. Please see the attachment.

Reviewer 2 Report

Khouchaf et al. analyzed the structure of amorphous silica depending on the ASR treatment time using HRTEM, EELS, e-PDF, and XRPD. Their analysis results are supporting the depolymerization of the Si-O-Si network structure and regarding the transition from Q4 structure to the other structures (Q1, Q2, and Q3) by alkali silica reaction (ASR). However, the structural transition by ASR can easily be expected, and I think just qualitative confirmation of the expectation is not very informative to the specialized interests to the readers of Materials. Moreover, there are too many formal errors in the manuscript. For example, Figure 1 has two (c). Therefore, I do not recommend this work to be published in Materials.

Author Response

(The authors gave the same response as above.)

Reviewer 3 Report

The manuscript entitled "Microstructure study of Amorphous Silica nanostructure using Electron microscopy and Electron Pair Distribution Function" contains a study of amorphous silica and modified ones (alkali silica reaction at different time intervals). The authors used various techniques, i.e. HRTEM, EELS, and XRPD. The authors stated that the investigated investigations allow to understand the silica nanostructure. This is a very good idea.

The obtained results are interesting. I have checked the manuscript and I would like to address a major revision before it is accepted by MATERIALS journal. Thus, the authors must address the following issues and comments before further processing.

  1. Tittle.

Title is unclear – it needs to be changed. My proposition: “Microstructure Study of Amorphous Silica Nanostructures using HRTEM, EELS, and XRPD”.

  1. Manuscript.

The article must be checked very carefully. I found a lot of minor typos.

  1. a) Absteact --> Abstract
  2. b) XRPD or PXRD??? See for example Abstract
  3. c) SiO2, see “2”: subscript or without subscript!!! See for example page 1, Introduction.
  4. d) page 2, line 78, [23-24] --> [23, 24]
  5. e) page 3, lines 104 and 118, duplicate explanation of ED abbreviation
  6. f) page 4, line 146, unnecessary initials of names, see also page 8, lines 260 and 263
  7. g) page 4, line 169, (Fig. 3.) --> (Fig. 3)
  8. h) page 4, line 171, 2b, 2c, 2d --> Fig. 2b-2d
  9. i) page 15, line 491, Theis --> Thesis.

etc.

  1. Introduction.

I suggest adding a paragraph about ReaxFF for silica nanostructures.

  1. Page 4, line 169.

Figure 10 after Fig. 3????

  1. Page 5, Fig. 2.
  2. a) Fig. 2c and 2d, old subtitles are visible.
  3. b) Fig. 2d, is the old inscription “20 nm”? Is Fig. 2d on the same scale as the others?
  4. c) What is the scale of enlarged images?
  5. d) What do the white lines mean in enlarged images?

  1. Page 5, lines 179-188.

The same information is duplicated in two paragraphs, i.e. “a diffuse peak maxima is located at 2θ = 21°” and “The broad peak for S1 phase 184 appears at 2θ = 21°”.

  1. Page 6, Fig. 3, y-axis.

Arbitrary units?

  1. Page 6, line 209.

“The ELNES spectra”?

  1. Pages 6 and 7, Figs. 4 and 5.

These figures should be merged.

  1. Pages 7 and 8, Figs. 6 and 7.

These figures should be merged.

  1. Page 7. Fig. 5 and the respective text in the manuscript.

This data does not follow a linear trend (For higher time values I expect a plateau). Please make appropriate statistical analysis confirming a linear trend or not!!!

  1. Page 8. Fig. 7 and the respective text in the manuscript.

This data does not follow a linear trend (For higher time values I expect a plateau). Please make appropriate statistical analysis confirming a linear trend or not!!!

  1. Page 8, Fig 8.

What about hydrogen bonds between adsorbed water molecules?

  1. Page 9, Fig. 9.
  2. a) y-axis. Arbitrary units?
  3. b) Why no measurements were made for S2 and S3 samples? Please take the respective measurements and discuss the results.

3) This figure is of very poor quality and different from others.

  1. Page 10, Fig. 10.
  2. a) y-axis. Arbitrary units?
  3. b) legend. “.gr” is unnecessary.

  1. Page 10, Tab. 1.

“(6 hours)”, “(168 hours)”, and “(312 hours)” are unnecessary. I suggest putting this table in Fig. 10.

  1. Page 11 lines 321-330 and Conclusions.

The following paragraph “As a conclusion, HRETM, XRD and e-PDF…” is unnecessary and duplicates Conclusions. Conclusions should be rebuilt.

  1. References.

I found a lot of minor typos and mistakes, for example,

[42] tittle?

[43] v. 29? (1994)?

[51] Theis?

[52] underlined tittel?

Author Response

(The authors gave the same response as above.)

Round 2

Reviewer 2 Report

The authors have improved the manuscript, and thus I recommend publication of the manuscript in Materials after addressing the following comments.

  1. Authors are arguing that there are only few examples in the literature where e-PDF in TEM. Then what is the difference between the authors’ results and the other few examples? Please make it clear and discuss differences in the Introduction.
  2. The authors’ main argument should be new finding not the use of several types of equipment. Please concentrate on what is the new findings from the structural analysis. For example, the following sentence is overemphasizing the use of several types of equipment.

“In this work, we used, for the first time, a combination of high-resolution transmission electron microscopy (HRTEM), X-ray powder diffraction (XRPD), electron energy loss spectroscopy (EELS), electron pair distribution function (e-PDF), and solid-state nuclear magnetic resonance (NMR) to study the structural evolution of silica nanostructures with different reaction times during the ASR process.”

What new finding which has not been available became available with the use of several types of equipment? The answer should be stated instead of overemphasizing. And the last paragraph in the Introduction should outline what will be discussed in the manuscript. But currently, the paragraph is too short and is not mentioning what will be discussed at all.

  1. The use of synonyms makes the manuscript hard to read. For example, silica, SiO2, silicone oxide are the same. Please unify the terms in the entire manuscript.
  2. Line 186. Figure 3c,d -> Figure 2c,d
  3. In the inset of Figure 2c-d, delete white lines and use an arrow or something other (dashes around the ring?) not covering the ring structure. The white line makes it difficult to determine what the ring structure is. Can the authors show where the rings in Figure 2c-d using are using arrows or can the authors indicate where those insets zoomed from?
  4. In Figure 4b and 5b, remove lines. Your data is only four dots and the others are not done.
  5. Is Figure 6 a new suggestion from authors or a reproduction of Dove et al.(48)? If it is a reproduction, authors should care about copyright.

Reviewer 3 Report

Comments to the Authors

The authors revised their manuscript well. This version can be accepted for publication after minor revision.

Page 2. I repeat question!!! Fig 2c and 2d. What do the white lines mean in the inset of this figure?

Pages 4 and 9, line 168, lines 305 and 306, Fig. 3. “4” should not be subscript. See text and careful check, eg. Q1 or Q1 (“1“ subscript). See also description of dashed vertical lines description in Fig. 3.

Page 6, Fig. 3. What does the vertical line mean? Please explain in the figure captions. It's a pity there are no colors like in Figs. 4 and 5.

Pages 6 and 7, Figs. 4(b) and 5(b). Lines connecting the points are unnecessary.
